# The Diversity of Gut Microbiota at Weaning Is Altered in Prolactin Receptor-Null Mice

**DOI:** 10.3390/nu15153447

**Published:** 2023-08-04

**Authors:** Ivan Luzardo-Ocampo, Ana Luisa Ocampo-Ruiz, José Luis Dena-Beltrán, Gonzalo Martínez de la Escalera, Carmen Clapp, Yazmín Macotela

**Affiliations:** Instituto de Neurobiología, Universidad Nacional Autónoma de México (UNAM), Querétaro 76230, Mexico; ivan.8907@gmail.com (I.L.-O.); analuisaocamporuiz@hotmail.com (A.L.O.-R.); jldena_271994@hotmail.com (J.L.D.-B.); gmel@unam.mx (G.M.d.l.E.); clapp@unam.mx (C.C.)

**Keywords:** bacterial diversity, gut microbiota, prolactin receptor, lactation, weaning

## Abstract

Maternal milk supports offspring development by providing microbiota, macronutrients, micronutrients, immune factors, and hormones. The hormone prolactin (PRL) is an important milk component with protective effects against metabolic diseases. Because maternal milk regulates microbiota composition and adequate microbiota protect against the development of metabolic diseases, we aimed to investigate whether PRL/PRL receptor signaling regulates gut microbiota composition in newborn mice at weaning. *16SrRNA* sequencing of feces and bioinformatics analysis was performed to evaluate gut microbiota in PRL receptor-null mice (*Prlr*-KO) at weaning (postnatal day 21). The normalized colon and cecal weights were higher and lower, respectively, in the *Prlr*-KO mice relative to the wild-type mice (*Prlr*-WT). Relative abundances (Simpson Evenness Index), phylogenetic diversity, and bacterial concentrations were lower in the *Prlr*-KO mice. Eleven bacteria species out of 470 differed between the *Prlr*-KO and *Prlr*-WT mice, with two genera (*Anaerotruncus* and *Lachnospiraceae*) related to metabolic disease development being the most common in the *Prlr*-KO mice. A higher metabolism of terpenoids and polyketides was predicted in the *Prlr*-KO mice compared to the *Prlr*-WT mice, and these metabolites had antimicrobial properties and were present in microbe-associated pathogenicity. We concluded that the absence of the PRL receptor altered gut microbiota, resulting in lower abundance and richness, which could contribute to metabolic disease development.

## 1. Introduction 

Trillions of microbial cells located in the intestinal compartment known as the “colonic microbiota” develop during childhood and adulthood and play important roles in promoting a host’s health [1]. Several factors, such as birth mode (vaginal or cesarean) and feeding method (breast milk or formula), influence the composition of gut microbiota [2]. Cesarean-born neonates have higher bacterial diversity than those born vaginally but are more prone to developing asthma, allergies, and obesity. This has been attributed, at least in part, to the gradual replacement of the *Bacteroides* genus, which helps regulate the immune system, by the *Firmicutes* genus in the first year of life [3]. Moreover, formula-fed children have less bacterial diversity and richness in the first 12–24 months than those fed breast milk, with lower levels of *Lactobacillus*, *Staphylococcus*, *Megasphaera*, and Actinobacteria [3]. 

Within the main microbial reservoirs that sustain an early neonate’s colonization, maternal milk is considered the second most abundant source after the mother’s areolar skin, contributing 8 × 10^5^ bacteria daily during lactation, particularly *Lactobacillus*, *Bifidobacterium*, *Staphylococcus* spp., and *Streptococcus* [4]. Milk components, such as milk oligosaccharides, serve as energy sources for select bacterial populations, which, in turn, produce short-chain fatty acids (SCFAs) and other metabolites that act as protectants against pathogens such as *Streptococcus pneumoniae* or *Campylobacter jejuni* [5]. 

However, not only are microbiota and macromolecules in milk delivered from a mother to an infant but also immune components, including immune cells, regulatory cells, and antibodies [6], as well as hormones such as prolactin (PRL), a protein that is known for its stimulatory effect on milk secretion and exerts a wide range of metabolic and immune actions [7]. Although PRL levels change depending on sex and physiopathological status, it has been recognized that the maintenance of a metabolically beneficial PRL level (HomeoFIT PRL: 7–100 µg/L) [8] could prevent metabolic disease development, whereas too-low and too-high PRL levels are associated with an increased prevalence of metabolic diseases [7,8]. During lactation, obesity is associated with reduced PRL action since the obese condition lowers the expression of PRL receptors (PRLRs) in mammary epithelial cells, hindering PRL signaling and causing a reduction in the production of milk components such as β–casein and α–lactalbumin, leading to lactation insufficiency and precocious mammary gland involution [9]. Additionally, high-fat diet (HFD) feeding in rats reduced PRL levels in maternal milk, and their pups consuming obesogenic and hypoprolactinemic milk developed exacerbated adiposity, fatty liver, and insulin resistance at weaning, whereas PRL administration to the HFD-fed mothers or directly to the pups ameliorated those metabolic alterations [10]. PRL treatment in HFD-fed lactating rat dams normalized mammary gland function and restored milk yields and PRL levels [10]. Therefore, PRL in maternal milk favors metabolic homeostasis in offspring, whereas a lack of adequate PRL actions derived from an obesogenic environment contribute to the development of metabolic diseases. 

Previous studies have suggested that communication between the endocrine system and microorganisms regulates a host’s hormonal homeostasis [11]. For instance, germ-free rats had 25% higher PRL levels than gnotobiotic (standard microbiota) animals [12]. Moreover, a 24 h SCFAs treatment could inhibit *Prl* expression in primary dairy cow anterior pituitary cells (DCAPCs) in vitro [13], supporting the idea that bacterial metabolites can influence endocrine factors. However, there are no reports about the relationship between the absence of PRLR signaling and gut microbiota composition in neonates, particularly during the weaning period, which is a critical timeframe for producing microbial adaptations that shape a neonate’s health and prevent the development of chronic conditions [14]. Altered microbiota have been implicated in several metabolic diseases such as cardiovascular diseases, obesity, and type 2 diabetes [15], and microbiota transplantation has been shown to reverse obesity and type 2 diabetes, and it has been used in the treatment of persistent and severe infections due to *C. difficile* [16].

Since maternal milk is a key regulator of gut microbiota composition and microbiota protect against the development of metabolic diseases, in this work, we investigated whether PRL/PRL receptor signaling regulates gut microbiota composition in newborn mice at the end of lactation. Our hypothesis was that the absence of PRLR induces changes in gut microbiota diversity and composition, promoting a microbial profile potentially linked to an increased risk of developing metabolic diseases. 

## 2. Materials and Methods

### 2.1. Animal Husbandry, Care, and Macroscopic Measurements

All animals were housed under standard laboratory conditions (12/12 h light–dark cycle, 20 °C, and 40–50% relative humidity). The animals were cared for by following the guidelines of the National Institutes of Health’s Guide for the Care and Use of Laboratory Animals. The experiments conducted were approved by the Bioethics Committee of the Institute of Neurobiology of the National Autonomous University of Mexico (ID: 075). All mice were fed a standard diet with pellets from Rodent Lab Chow 5001 (Purina, St. Louis, MO, USA). 

Male and female C57BL/6J *Prlr*
^+/−^ mice from The Jackson Laboratory were bred and maintained for several generations in the Vivarium of the Institute of Neurobiology of the National Autonomous University of Mexico (UNAM). After birth, the litter sizes were adjusted to 6–8 animals, and male and female *Prlr*
^+/+^ or *Prlr ^−/−^* pups (n: 15 pups/group) were maintained for 21 days until the lactation periods ended. The pups were then anesthetized by CO_2_ inhalation and euthanized by decapitation. The pups’ body weights were measured every two days starting from the fifth day after birth. The pups’ colons were excised and weighed. 

### 2.2. Fecal and Cecal DNA Extraction and Quality Control

Before conducting the extractions, to avoid contamination, the 21-day-old pups were placed in cages containing clean sawdust previously irradiated with UV for 15 min in a standard biosafety level 1 cabinet. The surfaces were cleaned with 70% ethanol. The fecal and cecal contents (200–300 mg) were extracted from the euthanized pups (n = 3/tube) and stored in DNA/RNA shield collection tubes (R1101, Zymo Research Corp., Irvine, CA, USA) at −80 °C. Genomic DNA was extracted using a ZymoBIOMICS DNA Miniprep kit (D4300, Zymo Research Corp.) in accordance with the manufacturer’s instructions. Once extracted, the DNA samples were quantified in a NanoDrop 1000 Spectrophotometer (Thermo Fisher, Waltham, MA, USA), and electrophoretic running was conducted in a 1.5% agarose gel for 30 min for 1:10 diluted samples to verify DNA integrity. 

### 2.3. Library Preparation, 16S rRNA Sequencing, and Diversity Index Analyses

A total of five samples per group (each sample being a pool of DNA from three mice) were diluted in sterile water with concentrations of 20 µg/µL. The samples were then processed and analyzed by the ZymoBIOMICS’ Targeted Metagenomics Sequencing (Zymo Research) service. The DNA samples were prepared for sequencing with a Quick-16S^TM^ NGS Library Pep Kit (Zymo Research) and a V3-V4 Primer Set. The sequencing library was prepared through real-time PCR reactions to quantify the pooled qPCR readings based on equal molarity. The library was then cleaned up with a Select-a-Size DNA Clean & Concentrator^TM^ (Zymo Research) and quantified with TapeStation^®^ (Agilent Technologies, Santa Clara, CA, USA) and Qubit^®^ (Thermo Fisher). The ZymoBIOMICS^®^ Microbial Community Standard (Zymo Research) was used as a positive control for each targeted library preparation. The final library was sequenced on an Illumina^®^ MiSeq^TM^ with a V3 reagent kit (600 cycles) and a 10% PhiX spike-in. The unique amplicon sequences were identified from the raw reads, and the chimeric sequences were removed using the DADA2 pipeline [17]. The taxonomy assignment was performed using Uclust from Qiime v. 1.9.1, following the Zymo Research 16S database. Composition visualization, α-diversity, and β-diversity analyses were also performed in Qiime v. 1.9.1 [18]. A quantitative real-time PCR was set up to quantify the absolute abundance, with a standard curve made with plasmid DNA containing one copy of the *16S* gene in 10-fold serial dilutions. The same primers used in the Targeted Library Preparation were used. The number of genome copies per DNA sample was calculated by dividing the gene copy number by an assumed number of gene copies per genome. The amount of DNA per microliter of sample was calculated using an assumed genome size of 4.64 × 10^6^ bp, the genome size of *Escherichia coli*. A two-dimensional principal coordinate analysis (PCoA) was conducted for the visual hierarchical clustering and community ordination using the web-based tool MicrobiomeAnalyst [19]. For the Phylogenetic Investigation of Communities by Reconstruction of Unobserved States (PICRUSt) to predict metabolic pathways [20], the whole genomes of the annotated species were searched using the Kyoto Encyclopedia of Genes and Genomes (KEGG) genome database (https://www.genome.jp/kegg/ko.html, accessed on 14 April 2023) and MicrobiomeAnalyst. 

### 2.4. Statistical Analysis

Except for the microbiota composition, where data are presented as the mean ± SDs of five values (each representing three animals), all other data are presented as the medians ± SDs of fifteen mice/group. After assessing the normality of the data using normal distribution, normal quantile plots, and a Shapiro–Wilk test, an analysis of variance (ANOVA) test followed by a post hoc Kruskal–Wallis multiple test were conducted to assess differences, with a cut-off *p*-value of <0.05 (or 0.01). GraphPad Prism v. 8.2 (Dotmatics, Boston, MA, USA) and MicrobiomeAnalyst were used to plot the data, and JMP v. 16.0 (SAS, Cary, NC, USA) was used to perform the statistical analyses. 

## 3. Results

### 3.1. Body Weight and Macroscopic Measurements

The macroscopic characteristics of the wild-type (*Prlr*
^+/+^, *Prlr*-WT) and knockout (*Prlr*
^−/−^, Prlr-KO) mice showed no differences in body weight evolution (Figure 1A) during lactation. However, the KO mice displayed higher normalized colon weights (*p* < 0.01) (Figure 1B) and reduced cecal weights (*p* < 0.05) (Figure 1C) compared to the WT mice.

### 3.2. General Microbial Diversity Analyses of the Weaned Prlr-WT and Prlr-KO Mice

The α-diversity in the bacterial compositions showed that both mouse genotypes shared 522 amplicon sequence variants (ASVs) (Figure 2), but the *Prlr*-KO pups displayed higher numbers of unshared ASVs (+2.96%) than the *Prlr*-WT animals (Figure 2A). Although no differences were found for the Shannon index values (Figure 2B), the *Prlr*
^+/+^ pups exhibited higher (*p* < 0.01) Simpson Evenness (Figure 2C) and phylogenetic diversity (Figure 2D) scores than the *Prlr*
^−/−^ pups. The Shannon index is a quantitative indicator of the number of different bacteria in the samples, indicating that higher Shannon index values equal increased community diversity [21]. On the other hand, the Simpson Evenness index indicates the probability that individuals will belong to the same species, and a high Simpson Evenness value indicates a less diverse bacterial community [21,22]. 

Although there were no differences in the genotype clusters for β-diversity (Figure 3) using a permutational analysis of variance (PERMANOVA) test, two differentiated clusters were shown for each genotype, where the *Prlr*-KO mice variation was contained within the *Prlr*-WT mice. 

### 3.3. Taxonomical Bacterial Composition

Next, we evaluated the relative abundances of the phylum (Figure 4A), class (Figure 4B), order (Figure 4C), and family (Figure 4D) distributions between the pups’ genotypes. Only the Betaproteobacteria class (Figure 4E), Burkhoderiales order (Figure 4F), *Alcaligenaceae* family (Figure 4G), and *Anaerotruncus* genus (Figure 4H) were different between the genotypes (*p* < 0.05). Overall, 9 phyla, 14 classes, 14 orders, and 23 families were found. 

Out of 469 species, 11 species were different between the genotypes (Figure 5). Although the identified species were unclassified according to the taxonomic database used, their known orders, families, or genera were placed accordingly. The *Prlr*
^+/+^ pups had greater abundances of species coming from the Bacteroidales and Clostridiales orders and the *Anaerotruncus* and *Ruminoclostridium* genera. On the other hand, the *Prlr*
^−/−^ pups had the lowest abundances of species from the *Lachnospiraceae* family and the *Roseburia* genus. 

### 3.4. PICRUSt Metabolic Prediction 

A PICRUSt metabolic prediction was performed based on the species classifications obtained from the pups’ genotypes (Figure 6). The prediction highlighted 11 probable metabolic pathways, but only the metabolism of terpenoids and polyketides was found to be different (*p* < 0.05) (Figure 6A). A PCoA analysis showed differences in the overall metabolic performances of the genotypes (*p* < 0.05), with the *Prlr*
^+/+^ samples being more alike compared to the *Prlr*
^−/−^ samples. 

## 4. Discussion

This research was intended to assess the impact of the absence of PRL/PRL receptor signaling on gut microbiota development in mice at weaning. The evaluation was conducted in a well-established PRLR knockout mouse model, originally created through gene targeting in 129svj [23] and later on C57BL/6 mice, where a 1.5 kb fragment of the targeting vector containing exon 5 was replaced with the similarly sized thymidine-neomycin (Tk-NO) cassette, resulting in an in-frame stop codon mutation [21]. The immunological characterization of *Prlr*-KO mice has indicated that these mice do not have a defective hematopoietic system [24], and the mice are capable of normal humoral and cell-mediated immune responses after exposure to T-independent/-dependent antigens [25,26]. Previous results from our research group have shown that *Prlr*-KO mice display slightly altered liver growth, with higher liver to body weight (LBW) ratios at 2 weeks of age but lower LBW ratios after 4 weeks of age compared to WT mice [22,23]. Adult *Prlr*-KO mice (16–18 weeks) do not show any differences in their visceral and subcutaneous adipose tissue weights or adipocyte areas compared to their WT counterparts. However, when challenged by high fat diet (HFD) feeding for 8 weeks, the *Prlr*-KO adult mice showed increased adiposity characterized by adipocyte hypertrophy, as well as exacerbated glucose intolerance and insulin resistance compared to HFD-fed WT mice [27]. Also, streptozotocin (STZ)-induced diabetes in *Prlr*-KO adult mice (5–7 weeks old) resulted in increased hyperglycemia and glucose intolerance and lower insulin levels than STZ-induced diabetic WT mice [24], whereas no abnormalities in glucose and insulin levels were observed in non-diabetic *Prlr*-KO mice compared to their WT counterparts. Thus, *Prlr*-KO mice show increased susceptibility to developing exacerbated metabolic diseases. Despite the many metabolic and phenotypic parameters that have been described in *Prlr*-KO mice, there are no reports about gut microbiota characterization in this mouse model. The rationale for studying the impact of PRL/PRL receptor signaling on gut microbiota at weaning was that maternal milk is a key regulator of gut microbiota composition and gut microbiota are critical for metabolic homeostasis in a host. Prolactin is a component of maternal milk regulating metabolism in offspring [10], and during lactation in rodents, maternal milk is the main source of prolactin as pituitary prolactin secretion (the primary source of circulating prolactin) starts approximately at weaning [28,29]. 

The time and order in which microbiota colonize the gut, as well as the nutrients/substrates they encounter, are critical and highly contribute to variations in microbiota between individuals [30]. Breast milk and alveolar skin provide an abundant number of microorganisms to a neonate, and the proportion of breast milk intake and its replacement with solid foods have significant impacts on microbiota diversity, events that have been proposed as the major drivers in the development of gut microbiota in human adults [2]. During weaning, *Lactobacillaceae* are gradually lost, followed by the expansion of *Clostridiaceae* [14], and these are critical bacterial families that prevent colonization by bacterial pathogens [31]. Since mouse microbiota in early life cannot protect a host against pathogen colonization, *Lactobacillaceae* presence is important as this bacterial family effectively and directly inhibits pathogens, contributes to barrier maintenance, and modulates a host’s immune system [32]. 

Along with microbiota and macro- or micro-nutrients, breast milk delivers immune and endocrine factors that contribute to a neonate’s nutrition and development. This supports the concept of breast milk as a biological system [33,34]. Among these endocrine factors, PRL is delivered via maternal milk from a mother to her offspring in humans [35] and rodents [10], and studies have suggested that it regulates neonatal metabolic homeostasis [10], which could involve the modulation of gut microbiota composition and diversity. Although the relationship between microbiota changes and PRL activity has been scarcely explored, variations in PRL levels are linked to metabolic changes, potentially implicating microbiota dysbiosis, and this is one of the mechanisms involved in the development of cardiovascular diseases, obesity, and type 2 diabetes [15]. Moreover, microbiota transplantation has been proven to reverse the severity of metabolic diseases [16]. Few reports have explored the association between endocrine factors and microbiota, and “microbial endocrinology” has been proposed as an emergent research area to study host–microbe interactions and how microbiota impact physiological processes, including endocrine and immune functions, which are critical to preventing the development of chronic NCDs [36]. Here, we evaluated the impact of the absence of PRLR-PRL signaling on the gut microbiota composition of mice at weaning. 

Macroscopically, both the WT and KO mice displayed similar body weight development throughout lactation. However, at weaning, the *Prlr*-KO mice showed significantly higher colon weights, which is an early indicator of a pro-inflammatory state [37]. Changes in microbiota diversity have been associated with stress during weaning due to ecological mechanisms after the change in nutrient supply or maturation of the immune system (e.g., maternal IgA replacement with endogenous IgA) [38]. Operational taxonomic units (OTUs) were originally used to group bacterial reads into clusters, considering the microbial sequence identity [39]. However, the use of denoising methods migrated to identify exact sequence variants or ASVs, allowing researchers to distinguish between the predicted “true” biological variations and those likely generated by sequencing errors, and even a single nucleotide variation is defined as a separated ASV [40]. Hence, the ASV variations in the tested animals indicated exact sequence variations interpreted as a differential bacterial composition for each mouse genotype, suggesting that the *Prlr*
^−/−^ mice had higher abundances of unique members of the bacterial communities, but this did not reveal the sizes of the populations (absolute abundances), as was observed by the lower Simpson Evenness and Phylogenetic diversity index scores of the *Prlr*
^−/−^ mice compared to those of the *Prlr*
^+/+^ mice [41]. Lower microbial diversity and richness are linked to adverse conditions such as increased gut permeability, NCDs (e.g., insulin resistance and obesity), and pro-inflammatory phenotypes [42]. Since PRL activity is absent in *Prlr*-KO animals, it could be feasible that an excess of milk-derived PRL in the colon could have influenced the bacterial populations as both the *Prlr*-WT and *Prlr*-KO mice were subjected to the same potential stressful conditions to their gut microbiota, but differential bacterial profiles could be observed at several taxonomical levels. Another possibility is that PRL is converted into smaller fragments in the intestinal milieu, such as vasoinhibins (Vi) [43], and that the effects observed in the *Prlr*-KO mice were the result of both a lack of PRLR signaling and an excess of Vi signaling. Excess Vi could be generated in the intestinal lumen derived from the cleavage of milk PRL or in the circulatory system of the pups resulting from the elevated PRL levels known to be present in the *Prlr*-KO animals. High PRL levels arose from the absent PRLRs, which normally exert a negative feedback loop, stimulating dopamine release and inhibiting PRL production and secretion [44]. 

The effect of the hormonal milieu on the gut bacterial compositions of newborn Wistar rats was recently evaluated, showing that daily oral administration of leptin and adiponectin decreased the levels of the Proteobacteria phylum and the *Blautia* genus [45]. However, in a past study, since a natural Proteobacteria decrease was presented along with intestinal maturation, the tested adipokines could enhance this process [46]. Moreover, it has been shown that leptin administration decreased *Sutterella* and increased *Clostridium* genera, while adiponectin decreased *Roseburia* and increased *Enterococcus* genera [45]. In another study, leptin concentrations in the maternal milk of women with obesity did not impact their neonates’ microbial diversity or compositions, but higher insulin concentrations in the mothers’ milk were correlated with increased taxonomic diversity, particularly for Gammaproteobacteria, and they were inversely correlated with *Streptococcaceae* [47]. It has also been reported that several bacterial genera could metabolize hormones such as progesterone or estradiol [48], and detectable plasmatic levels of progesterone (6–10 ng/mL) and 17β-estradiol (20–40 pg/mL) have been found in female rat pups at weaning [49], suggesting that colonic hormonal composition might influence bacterial growth based on their metabolism. There are no reports on the impact of PRL on gut microbiota, but germ-free rats were found to contain 25% more PRL plasma than gnotobiotic animals [50]. Moreover, bacterial families such as *Lactobacillaceae* (*Lactobacillus gasseri*, *L. crispatus,* and *L. jenesnii*), *Peptostreptococcus, Bifidobacteriaceae* (*Bifidobacterium longum*), and *Streptococcaceae* (*Streptococcus agalactiae* and *Streptococcus anginosus*) have been found to successfully grow in human follicular fluid [51], which contains several hormones, such as progesterone and PRL [52], advocating for potential interactions between microbiota and PRL. 

Regarding the mechanisms explaining the successful bacterial growth under hormone treatment, it has been found that the absence of estrogen receptor β signaling could differentially impact the overall abundances of bacterial phyla (e.g., Proteobacteria, Bacteroidetes, and Firmicutes) or orders (e.g., Lactobacillales) [53]. Moreover, the reported ability of *Clostridium scidens* to metabolize small traces of bile acids that easily escape from the small intestine into the cecum to produce adverse secondary bile acids, deoxycholic acid, lithocholic acid, and even glucocorticoids acting as signaling hormones in bacteria agrees with a proposed hormonal influence on bacteria [54]. Other reports have indicated that epinephrine and norepinephrine from a host activate the transcription of virulence genes and flagella regulation in enterohemorrhagic *E. coli.*, which involves the participation of a histidine kinase sensor located at the bacterial surface [55]. 

Betaproteobacteria (or its related order (Burkhoderiales)) has been linked to NCD development in human adults. For instance, Betaproteobacteria abundances are increased in humans with type 2 diabetes [56]. Five-year-old infants resulting from hypertensive pregnancies showed fewer *Alcaligenaceae* and *Coriobacteriaceae* families than those from normotensive pregnancies [57]. The *Anaerotruncus* genus, which we found to be increased in the *Prlr*-KO animals, is a butyrate-producing group of bacteria associated with obesity and correlates negatively with high carbohydrate-based diets but positively with total fat and the consumption of saturated fatty acids [58]. In rabbits, the *Anaerotruncus* genus was found to be negatively correlated with weaning weight [59].

Particularly for the differentially found species, the significant abundances of *Lachnospiraceae* species in the *Prlr*-WT mice could predict better intestinal health compared to the *Prlr*-KO mice since *Lachnospiraceae* are largely believed to be health-promoting species critical to maintaining colonic tissue due to their ability to produce SCFAs, the primary nutrition source for colonocytes [60], and known metabolites that decrease pro-inflammatory factors, protect the colonic mucosa, and inhibit NLRP3 inflammasome activation and reactive oxygen species (ROS) production [61]. Increased *Lachnospiraceae* abundance in 2–9–week-old children is considered one of the collective microbiota characteristics of appropriate growth after birth, together with augmented microbial diversity, higher abundances of *Streptococcus* and strictly aerobic taxa, and decreases in *Staphylococcus* and *Enterobacteriaceae* abundances [62]. 

The predicted functional analysis was consistent with maternal milk harboring bacteria linked to carbohydrates, amino acids, and energy metabolism [63]. Although reports are scarce, a recent article indicated that the early exposure of neonates to elevated leptin and insulin concentrations from the maternal milk of women with obesity could impact the neonates’ metagenomic profiles as high leptin concentrations are inversely correlated with bacterial amino acid, carbohydrate, vitamin, and amino acid metabolism [64]. Products such as polyketides, alkaloids, and terpenoids are only derived from plants and can be formed by microorganisms [65]. Most terpenoids are terpene derivatives not encoded by microbiome genomes but represent microbial metabolites from dietary products or bile acid derivatives, and some of them are the result of the activities of oxidizing enzymes, such as terpene cyclases or synthases, and the addition of carbohydrates, amino acids, and fatty acid chains into polycyclic terpene backbones [66,67]. Some terpenes are widely synthesized by Proteobacteria, Actinobacteria, Firmicutes, and Bacteroidetes and display a wide range of biological properties, but little information regarding their colonic biosynthesis and effects has been reported [68]. On the other hand, polyketides are secondary metabolites produced mainly by Actinobacteria, Proteobacteria, Bacteroidetes, and Firmicutes, and they exhibit antimicrobial properties against select populations [69]. 

## 5. Conclusions

The results obtained in this research suggested that the absence of PRLR signaling could promote a higher abundance of gut microbiota potentially linked to NCDs during weaning. Both the *Prlr*-WT and *Prlr*-KO weaned mice shared similarities in bacterial diversity, taxonomic compositions, and metabolic functionalities, but the *Prlr*-KO mice displayed differential bacterial species that could predispose the mice to adverse disease conditions. Although it is challenging to establish a mechanism of action based on the results presented in this research, the fact that a lack of prolactin receptor signaling results in altered gut microbiota at a key developmental timepoint opens new research avenues that merit additional research. Moreover, these results supported the hypothesis that an altered microbiota profile in *Prlr*-KO mice contributes to their observed susceptibility to developing aggravated metabolic diseases.

## Figures and Tables

**Figure 1 nutrients-15-03447-f001:**
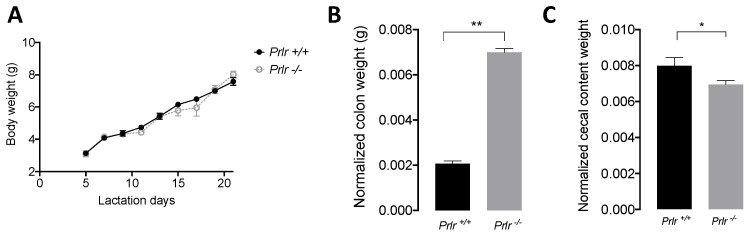
Macroscopic measurements of the pups. (**A**) Body weight evolution during lactation. (**B**) Normalized colon weights to body weights. (**C**) Normalized cecal content weights to body weights. The data are presented as the mean ± SDs of 15 mice. The asterisks indicate significant differences (* *p* < 0.05 and ** *p* < 0.01) according to the Kruskal–Wallis test.

**Figure 2 nutrients-15-03447-f002:**
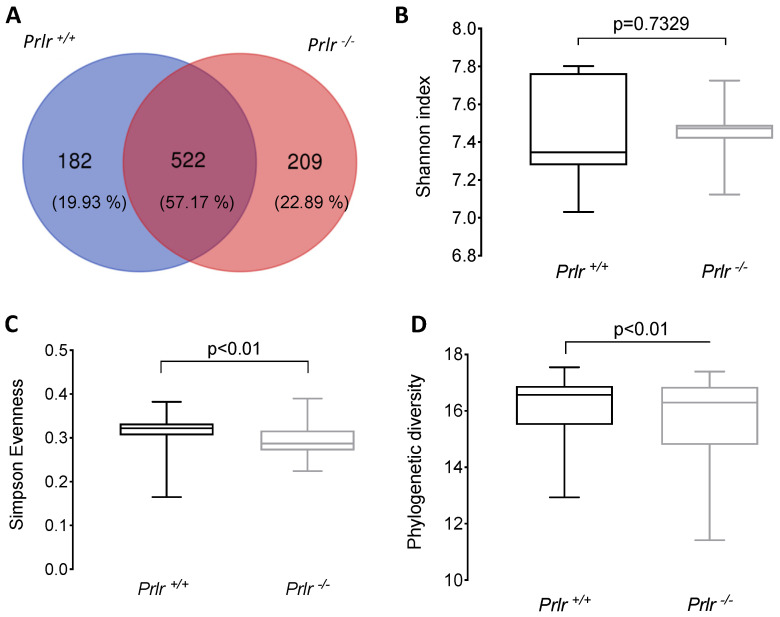
Alpha-diversity of the bacterial compositions of the weaned 21-day-old pups. (**A**) Venn diagram of the shared and unshared amplicon sequence variants (ASVs) between the groups; (**B**) Shannon index; (**C**) Simpson evenness; and (**D**) Faith’s phylogenetic diversity. The data (**B**–**D**) are presented as the mean ± SD of five samples (three mice each). Differences were assessed through a Kruskal–Wallis test.

**Figure 3 nutrients-15-03447-f003:**
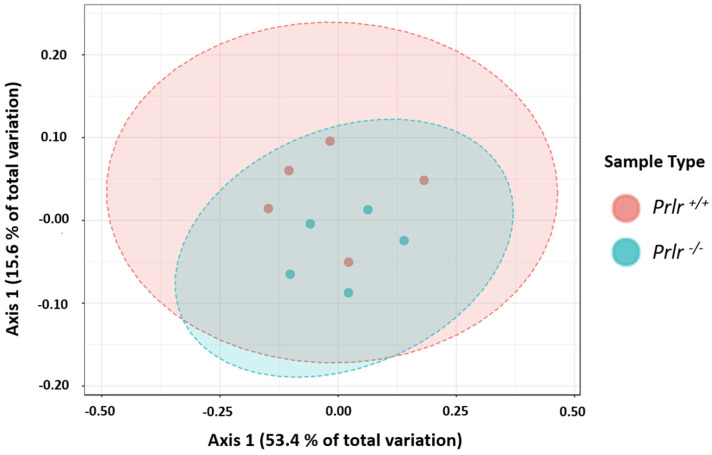
Bray–Curtis distance matrix PCoA plot. Each dot represents pooled data from three mice (n = 15/group). The analysis was conducted considering the species diversity, and the data were not significant (*p* = 0.075) based on a permutational analysis of variance (PERMANOVA) analysis (*p* > 0.05).

**Figure 4 nutrients-15-03447-f004:**
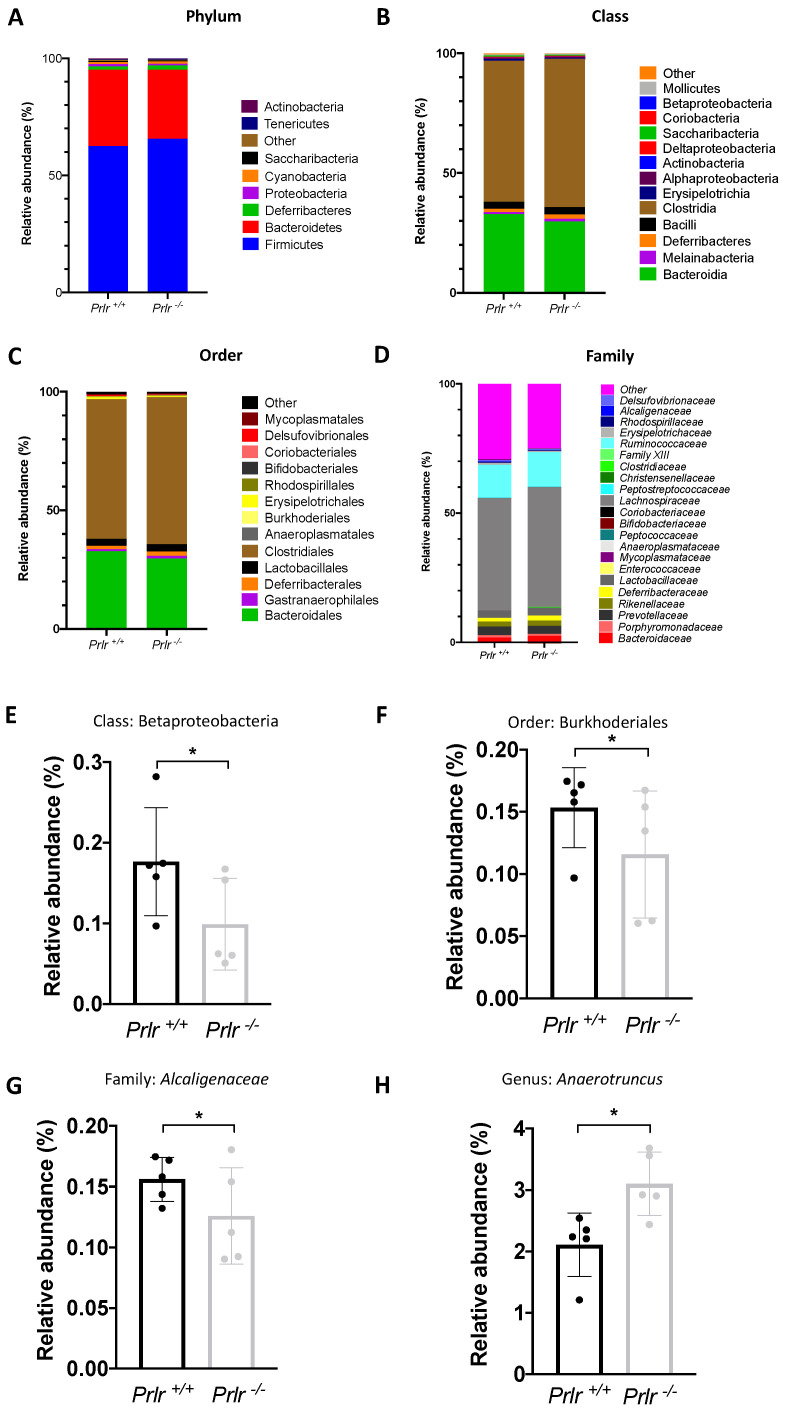
Taxonomic bacterial composition of the groups (21-day-old mice). Relative abundances of phylum (**A**), class (**B**), order (**C**), and family (**D**). Significantly different (*p* < 0.05) classes (**E**), orders (**F**), families (**G**), and genera (**H**). The data are presented as the mean ± SD of five samples (three mice each). Differences were assessed through a Kruskal–Wallis test (* *p* < 0.05).

**Figure 5 nutrients-15-03447-f005:**
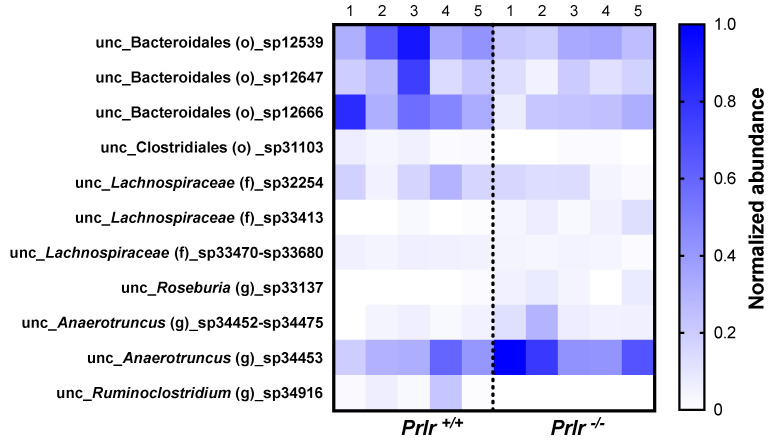
Significantly different bacterial species between the 21-day-old pups in each group (five samples from each group were taken, each representing three mice). The values are presented in normalized abundances after a min-max normalization [(sample − min)/(max − min)]. F, family; G, genus; O, order; UNC, unclassified.

**Figure 6 nutrients-15-03447-f006:**
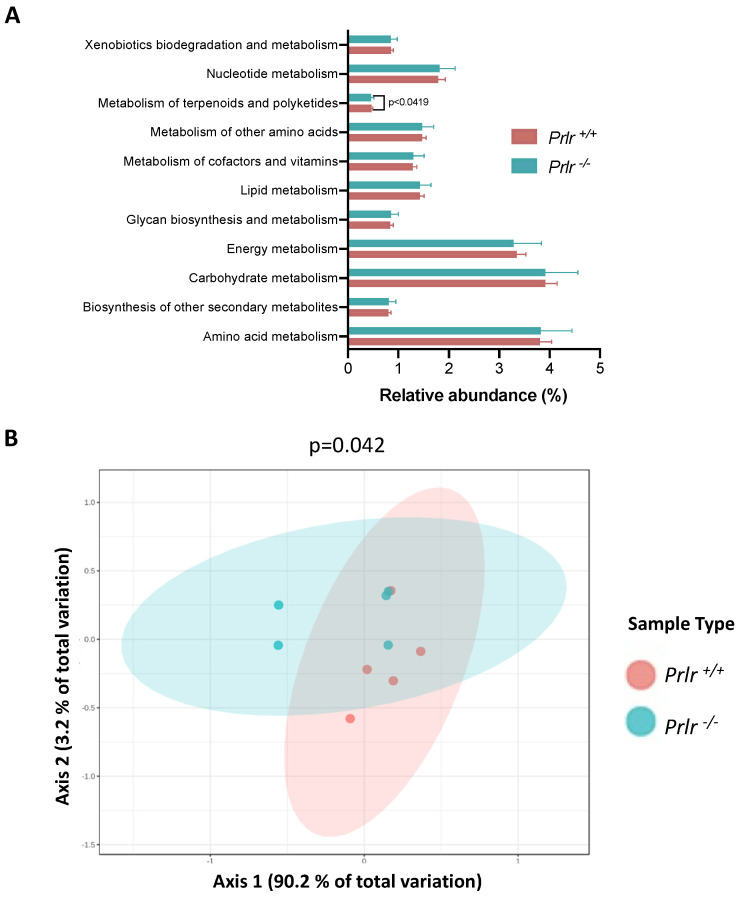
PICRUSt metabolic prediction based on species compositions from the 21-day-old weaned mice. (**A**) Relative metabolic abundances for each group. (**B**) PCoA analysis of the metabolic performances. The data in (**A**) are presented as the mean ± SD of 469 identified species. The differences in (**A**) were evaluated through a Kruskal–Wallis test (*p* < 0.05). The differences in (**B**) were assessed using a PERMANOVA test (*p* < 0.05). Each dot in (**B**) represents three mice.

## Data Availability

The data are available upon reasonable request.

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
