# Peer review of "The Diversity of Gut Microbiota at Weaning Is Altered in Prolactin Receptor-Null Mice"

_nutrients, 2023, doi:10.3390/nu15153447_

Round 1
Reviewer 1 Report
This manuscript explored the effects of PRL receptor knockout (Prlr KO) on pre weaning body weight, colon and cecal weights, and fecal microbiota in mice. The experimental purpose of the article is not clear. The introduction introduces the impact of PRL levels in breast milk on offspring, but the experimental design is to evaluate the changes in body weight and intestinal microbiota of Prlr-KO mice before weaning. The phenotypic data of the article is insufficient, such as neither the establishment of the Prlr-KO mice model had been estabished, nor the PRL levels, metabolic indices (fat liver, insulin resistance), and immune related indicators of mice had been measured. It is difficult to draw sound conclusions based solely on the results of 16SrRNA sequencing of feces.
Author Response
Reviewer 1
- Reviewer: This manuscript explored the effects of PRL receptor knockout (Prlr-KO) on pre-weaning body weight, colon and cecal weights, and fecal microbiota in mice. The experimental purpose of the article is not clear. The introduction introduces the impact of PRL levels in breast milk on offspring, but the experimental design is to evaluate the changes in body weight and intestinal microbiota of Prlr-KO mice before weaning.
- Author’s response: We appreciate the reviewer’s comment and acknowledge the need to clarify the purpose of the article. This research was intended to elucidate if the absence of prolactin receptor signaling could influence gut microbiota composition in vivo in mice at weaning. Hence, a very well-characterized prolactin receptor knock-out model was used to ensure there was no prolactin signaling activity in these mice. The rationale for studying the impact of prolactin/prolactin receptor signaling on gut microbiota at weaning is that maternal milk is a key regulator of gut microbiota composition and gut microbiota is critical for metabolic homeostasis of the host. Prolactin is a component of maternal milk regulating metabolism in the offspring (de los Rios et al., 2018), and during lactation in rodents, maternal milk is the main source of prolactin, as pituitary prolactin secretion (the primary source of circulating prolactin) starts around weaning (Ben-Jonathan et al., 2008; Moreno-Carranza et al., 2018). Therefore, we explored whether the effects of PRL/PRL receptors on lactating animals involve the regulation of gut microbiota composition.
References
Ben-Jonathan, N., LaPensee, C. R., & LaPensee, E. W. (2008). What can we learn from rodents about prolactin in humans? Endocrine Reviews, 29(1), 1–41. https://doi.org/10.1210/er.2007-0017
de los Ríos, E. A., Ruiz-Herrera, X., Tinoco-Pantoja, V., López-Barrera, F., Escalera, G. M., Clapp, C., & Macotela, Y. (2018). Impaired prolactin actions mediate altered offspring metabolism induced by maternal high‐fat feeding during lactation. The FASEB Journal, 32(6), 3457–3470. https://doi.org/10.1096/fj.201701154R
Moreno-Carranza, B., Bravo-Manríquez, M., Baez, A., Ledesma-Colunga, M. G., Ruiz-Herrera, X., Reyes-Ortega, P., de los Ríos, E. A., Macotela, Y., Martínez de la Escalera, G., & Clapp, C. (2018). Prolactin regulates liver growth during postnatal development in mice. American Journal of Physiology-Regulatory, Integrative and Comparative Physiology, 314(6), R902–R908. https://doi.org/10.1152/ajpregu.00003.2018
Revised manuscript:
- Page 1, Lines 13-17: “The hormone prolactin (PRL) is an important milk component with protective effects against metabolic diseases. Because maternal milk regulates microbiota composition and an adequate microbiota protects against the development of metabolic diseases, we aimed to investigate whether PRL/PRL receptor signaling regulates gut microbiota composition in newborn mice at weaning.”
- Page 2, Lines 83-86: “Since maternal milk is a key regulator of gut microbiota composition, and microbiota protects against the development of metabolic diseases, in this work, we investigated whether PRL/PRL receptor signaling regulates gut microbiota composition in newborn mice at the end of lactation.”
- Page 9, Lines 252-260: “Despite several metabolic and phenotypic parameters have been described in the Prlr-KO mice, there are no reports about gut microbiota characterization in this mouse model. The rationale for studying the impact of PRL/PRL receptor signaling on gut microbiota at weaning is that maternal milk is a key regulator of gut microbiota composition and gut microbiota is critical for metabolic homeostasis of the host. Prolactin is a component of maternal milk regulating metabolism in the offspring [10], and during lactation in rodents, maternal milk is the main source of prolactin, as pituitary prolactin secretion (the primary source of circulating prolactin) starts around weaning [22,25].”
- Reviewer: The phenotypic data of the article is insufficient, such as neither the establishment of the Prlr-KO mice model had been established, nor the PRL levels, metabolic indices (fat liver, insulin resistance), and immune-related indicators of mice had been measured
- Authors’ response: Thanks for the comment. The Prlr-KO mouse model, is a well-established model that our research group and other groups in the world have used for several years. However, following the reviewer’s pertinent request we have added more information about the model and expanded on the observed phenotype of the mice. Please refer to the revised manuscript.
Revised manuscript
- Pages 8-9, Lines 231-252: This research was intended to assess the impact of the absence of PRL/PRL receptor signaling in gut microbiota development on mice at weaning. The evaluation was conducted in a well-established PRLR knockout mouse model, originally created through gene targeting in 129svj [23] and later on C57BL/6 mice, where a 1.5 kb fragment of the targeting vector containing exon 5 was replaced with the similarly sized thymidine-neomycin (Tk-NO) cassette, resulting in an in-frame stop codon mutation [21]. Immunological characterization of the Prlr-KO mice has indicated that these mice do not have a defective hematopoietic system [24] and mice are capable of normal humoral and cell-mediated immune responses after exposure to T-independent/dependent antigens [25,26]. Previous results from our research group have shown that Prlr-KO mice display slightly altered liver growth, with higher liver to body weight (LBW) ratio at 2 weeks of age but lower LBW ratio after 4 weeks of age, compared to WT mice [22,23]. Adult Prlr-KO mice (16-18 weeks) do not show any differences in the visceral and subcutaneous adipose tissue weight or adipocyte area, compared to their WT counterparts. However, when challenged by a high fat diet (HFD) feeding for 8 weeks, Prlr-KO adult mice showed increased adiposity, characterized by adipocyte hypertrophy, and exacerbated glucose intolerance and insulin resistance compared to HFD-fed WT mice [27]. Also, streptozotocin (STZ)-induced diabetes in Prlr-KO adult mice (5-7 weeks old) resulted in increased hyperglycemia and glucose intolerance (p<0.05), and lower insulin levels (p>0.05) than STZ-induced diabetic WT mice [24], whereas no abnormalities in glucose and insulin levels were observed in non-diabetic Prlr-KO mice compared to their WT counterparts. Thus, Prlr-KO mice show increased susceptibility to develop exacerbated metabolic diseases.”
- Reviewer: It is difficult to draw sound conclusions based solely on the results of 16SrRNA sequencing of feces.
- Authors’ response: The reviewer has a valid point. However, this is the first report about the impact of the absence of prolactin/prolactin receptor signaling in the modulation of gut microbiota. Therefore, we believe our results open an interesting new avenue of research. However, we have modified our conclusions to acknowledge that additional research is still needed to fully understand the implications of the observed results. Also, following the reviewer´s suggestion, we have added more information regarding the metabolic phenotype of the mice, which supports the idea that an altered microbiota profile in Prlr-KO mice contributes to their susceptibility to develop aggravated metabolic diseases. Please refer to the revised manuscript.
Revised manuscript:
- Page 11, Lines 394-400: “Although it is challenging to establish a mechanism of action based on the results presented in this research, the fact that lack of prolactin receptor signaling derives in altered gut microbiota at a key developmental time point, opens new research avenues that merit additional research. Moreover, these results support the hypothesis that an altered microbiota profile in Prlr-KO mice contribute to their observed susceptibility to develop aggravated metabolic diseases.”

Reviewer 2 Report
The paper by Dr. Ivan Luzardo-Ocampo describes PRL signaling is required for maintaining the gut microbial diversity at weaning. PRL is known as preventing NCD development. This study might shed light on relationship of PRL signaling in gut and NCD at weaning but some of the methods need to be clarified.
1. The authors described abundance of species from Lachnospiraceae familty and the Roseburia were increased in Prlr-KO(line 202-203) but they are increased in WT in discussion (line 318). Which is true? Futhermore it seems there are not so much difference. How much is p value?
2. It is unclear what phenotype of KO at weaning or growing are.
3. The authors mainly focus on relationship of microbiota and metabolic disease in discussion. NCD is included with some disease such as cancer and chronic respiratory disease. Please clarify why the authors implicate alternation of KO mice microbiota in NCD.
Author Response
Reviewer 2
- Reviewer: The paper by Dr. Ivan Luzardo-Ocampo describes PRL signaling is required for maintaining the gut microbial diversity at weaning. PRL is known as preventing NCD development. This study might shed light on relationship of PRL signaling in gut and NCD at weaning but some of the methods need to be clarified.
- Authors’ response: We appreciate the reviewer’s comments.
- Reviewer: The authors described abundance of species from Lachnospiraceae family and the Roseburia were increased in Prlr-KO (line 202-203) but they are increased in WT in discussion (line 318). Which is true? Futhermore, it seems there are not so much difference. How much is p value?
- Authors’ response: We apologize for the mistaken information and thank the reviewer for pointing this out. Information in lines 202-203 was incorrect and has been modified as follows:
- Page 7, Lines 210-211: “Prlr -/- pups had the lowest abundance of species from the Lachnospiraceae family and the Roseburia
The p-values are presented as follows
Bacterial species |
p-values |
unc_Lachnospiraceae (f)_sp32254 |
0.0472 |
unc_Lachnospiraceae (f)_sp33413 |
0.0082 |
unc_Lachnospiraceae (f)_sp33470-sp33680 |
0.0283 |
unc_Roseburia (g)_sp33137 |
0.0343 |
Comparisons were conducted using a multiple-range non-parametric test (Kruskal-Wallis test) between WT and KO animals.
- Reviewer: It is unclear what phenotype of KO at weaning or growing are.
- Authors’ response: Animals were tested at postnatal day 21, coinciding with the end of lactation (both WT and KO). Their phenotype has been widely characterized by our research group. We have added more information about this animal model regarding their metabolic phenotype. Please refer to the revised manuscript.
- Pages 8-9, Lines 231-252: This research was intended to assess the impact of the absence of PRL signaling in gut microbiota development on mice at weaning. The evaluation was conducted in a well-established PRLR knockout mouse model, originally created through gene targeting in 129svj [23] and later on C57BL/6 mice, where a 1.5 kb fragment of the targeting vector containing exon 5 was replaced with the similarly sized thymidine-neomycin (Tk-NO) cassette, resulting in an in-frame stop codon mutation [21]. Immunological characterization of the Prlr-KO mice has indicated that these mice does not have a defective hematopoietic system [24] and mice are capable of normal humoral and cell-mediated immune response after exposure to T-independent/dependent antigens [25,26]. Previous results from our research group have shown that Prlr-KO mice display slightly altered liver growth, with higher liver to body weight (LBW) ratio at 2 weeks of age but lower LBW ratio after 4 weeks of age, compared to WT mice [22,23]. Adult Prlr-KO mice (16-18 weeks) have not shown any differences in the visceral adipose tissue (VAT) and subcutaneous adipose tissue (SAT) weight or adipocyte area, compared to their WT counterparts. However, when challenged by a high fat diet (HFD) feeding, Prlr-KO adult mice showed increased adiposity, characterized by adipocyte hypertrophy, and exacerbated glucose intolerance and insulin resistance compared to HFD-fed WT mice [27]. Also, streptozotocin (STZ)-induced diabetes in Prlr-KO adult mice (5-7 weeks old) resulted in increased hyperglycemia and glucose intolerance (p<0.05), and lower insulin levels (p>0.05) than STZ-induced diabetic WT mice [24], whereas no abnormalities in glucose and insulin levels were observed in non-diabetic Prlr-KO mice compared to their WT counterparts. Thus, Prlr-KO mice show increased susceptibility to develop exacerbated metabolic diseases.”
- Reviewer: The authors mainly focus on relationship of microbiota and metabolic disease in discussion. NCD is included with some disease such as cancer and chronic respiratory disease. Please clarify why the authors implicate alternation of KO mice microbiota in NCD.
- Authors’ response: The reviewer is right in that NCD is a very wide term that includes not only metabolic diseases but also cancer and respiratory diseases. To be more specific, we have changed NCDs to “metabolic diseases” throughout the paper. We have clarified that there is evidence that PRL activity in lactating rodents prevents metabolic alterations derived from obesity, such as fatty liver, insulin resistance, and excessive adiposity (de los Rios et al 2018), and that in humans, low PRL levels associate with increased prevalence of those metabolic diseases and others such as type 2 diabetes (Macotela et al 2020, 2022). Also, altered microbiota has been implicated in several metabolic diseases such as cardiovascular diseases, obesity and type 2 diabetes (Noce et al., 2019), and microbiota transplantation has been shown to reverse obesity, type 2 diabetes, or being used in the treatment of persistent and severe infections due to difficile (Choi & Cho, 2016). Therefore, in this work, we aimed to study whether the actions of prolactin/prolactin receptors in lactating animals, could involve the regulation of the gut microbiota. We have also acknowledged that the results presented here do not allow us to dissect the mechanisms linking prolactin metabolic beneficial actions with the regulation of the microbiota, however, these results open new avenues to explore such mechanisms in detail. We hope this further explanation clarified the ideas and rational of the study and results. We have added an explanation for this rationale in the revised manuscript.
Revised manuscript:
- Page 1, Lines 13-17: “The hormone prolactin (PRL) is an important milk component with protective effects against metabolic diseases. Because maternal milk regulates microbiota composition and an adequate microbiota protects against the development of metabolic diseases, we aimed to investigate whether PRL/PRL receptor signaling regulates gut microbiota composition in newborn mice at weaning.”
- Page 2, Lines 53-69: “Although PRL levels change depending on sex and physiopathological status, it has been recognized that the maintenance of a metabolically beneficial PRL level (HomeoFIT PRL: 7-100 µg/L) [8] could prevent metabolic diseases Whereas too low and too high PRL levels associate with increased prevalence of metabolic diseases [7,8]. During lactation, obesity is associated with reduced PRL action since the obese condition lowers the expression of PRL receptors (PRLRs) in mammary epithelial cells, hindering PRL signaling and causing a reduction in the production of milk components like b-casein and a-lactalbumin, leading to lactation insufficiency and precocious mammary gland involution [9]. Additionally, high-fat diet (HFD) feeding in rats reduces PRL levels in maternal milk, and their pups consuming obesogenic and hypoprolactinemic milk develop exacerbated adiposity, fatty liver, and insulin resistance at weaning, whereas PRL administration to the HFD-fed mothers or directly to the pups ameliorates those metabolic alterations [10]. PRL treatment in HFD-fed lactating rat dams normalized mammary gland function and restored milk yield and PRL levels [10]. Therefore, PRL in maternal milk favors metabolic homeostasis in the offspring, whereas lack of adequate PRL actions derived from an obesogenic environment contribute to the development of metabolic diseases.”
- Page 2, Lines 79-88: Altered microbiota has been implicated in several metabolic diseases such as cardiovascular diseases, obesity, and type 2 diabetes (Noce et al., 2019), and microbiota transplantation has been shown to reverse obesity, type 2 diabetes, or being used in the treatment of persistent and severe infections due to difficile (Choi & Cho, 2016).
Since maternal milk is a key regulator of gut microbiota composition, and microbiota protects against the development of metabolic diseases, in this work we investigated whether PRL/PRL receptor signaling regulates gut microbiota composition in newborn mice at the end of lactation. Our hypothesis was that the absence of PRLR induces changes in gut microbiota diversity and composition, promoting a microbial profile potentially linked to an increased risk of developing metabolic diseases.
- Page 9, Lines 278-283: “Although the relationship between microbiota changes and PRL activity has been scarcely explored, variations in PRL levels are linked to metabolic changes potentially implicating microbiota dysbiosis, as this is one of the reasons involved in the development of cardiovascular diseases, obesity and type 2 diabetes [15]. Moreover, microbiota transplantation has proven to reverse the severity of metabolic diseases [16].”
- Page 11, Lines 394-400: “Although it is challenging to establish a mechanism of action based on the results presented in this research, the fact that lack of prolactin receptor signaling derives in altered gut microbiota at a key developmental time point, opens new research avenues that merit additional research. Moreover, these results support the hypothesis that an altered microbiota profile in Prlr-KO mice contribute to their observed susceptibility to develop aggravated metabolic diseases.”

Round 2
Reviewer 2 Report
The review is written better than last time. I have no comments.